# Possibilities of Manufacturing Products from Cermet Compositions Using Nanoscale Powders by Additive Manufacturing Methods

**DOI:** 10.3390/ma12203425

**Published:** 2019-10-19

**Authors:** Sergei Grigoriev, Tatiana Tarasova, Andrey Gusarov, Roman Khmyrov, Sergei Egorov

**Affiliations:** Moscow State University of Technology “STANKIN”, Moscow 127055, Russia

**Keywords:** additive manufacturing, selective laser melting, cermet, WC–Co, nanopowder, hard alloy

## Abstract

Complicated wear-resistant parts made by selective laser melting (SLM) of powder material based on compositions of metal and ceramics can be widely used in mining, oil engineering, and other precision engineering industries. Ceramic–metal compositions were made using nanoscale powders by powder metallurgy methods. Optimal regimes were found for the SLM method. Chemical and phase composition, fracture toughness, and wear resistance of the obtained materials were determined. The wear rate of samples from 94 wt% tungsten carbide (WC) and 6 wt% cobalt (Co) was 1.3 times lower than that of a sample from BK6 obtained by the conventional methods. The hardness of obtained samples 2500 HV was 1.6 times higher than that of a sample from BK6 obtained by the traditional method (1550 HV).

## 1. Introduction

Additive manufacturing is the common name for a family of layer-by-layer production technologies using electronic CAD-models. The principle of additive manufacturing is to create functional products with the help of layer-by-layer addition of material by depositing or spraying powder and by adding a liquid polymer or composite [1,2]. 

One of the most promising technologies for additive manufacturing is the technology of selective laser melting (SLM) (Powder Bed Fusion—“melting the material in a preformed layer”) because it has a number of fundamental advantages: it is waste-free, has versatility, and has the ability to manufacture high-precision complex parts that are not inferior, and which are sometimes even superior in their physicomechanical characteristics than the parts obtained by traditional shaping. This technology can reduce the manufacturing time and cost of complex parts in single and small batch production due to the lack of a stage for creating a special tool and a reduction in the number of technological stages [3,4,5,6].

Selective laser melting uses a wide range of materials. Using such materials as aluminum alloys, corrosion-resistant steels, and cobalt-based alloys it is possible to obtain a high complexity of physicomechanical properties of products made by the SLM method [7,8,9,10,11,12]. 

When applying SLM to brittle materials, the part often cracks already in the manufacturing process. Significant temperature gradients occur during laser heating. Inhomogeneous thermal fields and thermal expansions generate thermomechanical stresses. Stresses can exceed the mechanical strength of the material, so the material processed by the laser should be resistant to the thermal shocks. Up to now, there have been no successful attempts to manufacture parts from ceramic materials comparable in mechanical properties to parts obtained by classical technology. However, the interest in SLM for ceramics is quite high [13,14,15].

One of the promising materials is metal-based composite materials with the addition of ceramic inclusions. Hard alloys are a classic example of cermet materials, which are well studied and widely used. Hard alloys are based on a matrix of alloys of elements of the iron group hardened by carbides of refractory metals [16,17,18,19,20]. The mechanical properties of hard alloys are significantly higher than those of traditional alloys. By combining various matrices and reinforcing components, it is possible to obtain composite materials with desired properties, which allows us to solve the problem of optimizing the structures to obtain required characteristics. Thus, when using a ductile matrix and hard reinforcing inclusions, two opposite properties necessary for structural materials are combined: high tensile strength and sufficient fracture toughness. At present, hard alloys are mainly produced by powder metallurgy methods, the complexity of which inhibits their wider application. Metal matrix composites with a high content of hardening phases are poorly formed and processed by traditional methods, which inhibits their widespread use. Layer-by-layer molding from powders by selective laser melting could solve this problem. 

Immediately after the advent of SLM technology, interest arose in obtaining by this method compositions with a metal matrix. For example, Laoui et al. in 1999 studied the WC–Co system [21], and Xiao et al. studied the TiC-invar system in 2000 [22].

Further, the list of tested composite materials for SLM expanded; however, the obtained materials were of unsatisfactory quality.

Examples of use of composite materials with a metal matrix are described in the literature more often for laser surfacing than for SLM. In laser surfacing, the distribution of temperature and cooling rates are similar to those for SLM; therefore, they require consideration. In [23], coatings were obtained by laser surfacing of a mixture of tungsten carbide and nickel alloy powders. The volumetric concentration of tungsten carbides was about 40%. The metal matrix consisted of a plastic solid solution based on nickel and a solid and brittle eutectic mixture. The ratio of the plastic and solid phases in the matrix was very different depending on the chromium content in the nickel alloy. At a high chromium content (12–14% by weight), when brittle eutectics prevailed in the matrix structure, the coating cracked during deposition. Cracks extended across the coating from edge to edge and were perpendicular to the direction of laser scanning. At a low chromium content (6–8% by weight), when the plastic nickel phase prevailed in the matrix structure, no cracks were observed [23].

Thus, the authors concluded that when manufacturing composite materials by laser welding, it is important to obtain a plastic matrix so that the material does not crack.

Titanium carbide is also widely used as a hardening phase in laser surfacing. Its density is about four times lower than that of tungsten carbide, which is essential for use in the aerospace industry. Candel et al. [24] obtained crack free coatings using a TiC–Ti composition with a carbide content of 30% and 60%. In the structure, primary and secondary carbides were also observed. Leunda et al. [25] achieved good results in hardening steel with vanadium carbide, introducing it up to 10% by weight of laser surfacing.

A promising direction for improving the mechanical properties of carbides is reducing the grain size [26]. For this purpose, mechanical alloying of the powders of the starting components is used. Methods have been developed for producing a mechanically nanostructured alloy from micron-sized starting powders [27].

SLM is not fundamentally limited by the complexity of the geometric shape and refractoriness of the material. Moreover, high cooling rates of the order of 10^6^ K/s, specific for this process, often provide a fine-grained submicron or nanoscale structure with increased strength and wear resistance. Complicated parts made by SLM of powder material based on compositions of metal and ceramics can be widely used in mining, oil engineering and other precision engineering industries. Therefore, the technology of manufacturing from metal powder compositions by selective laser melting is an important scientific and technological task.

The aim of this work is researching and developing wear-resistant ceramic–metal compositions of the WC–Co system obtained by SLM using nanosized powders.

## 2. Materials and Methods

As an experimental material for selective laser melting, the WC–Co system was chosen. To obtain powders of composite materials, we used cobalt metal powder and ceramic powder of tungsten carbide: nanoscale and micron powders of cobalt and WC. The substrates were carbide plates of grade BK20 GOST 19106-73 (Co—20 wt%; WC—80 wt%), production, Russia, Kirovgrad, Kirovgrad hard alloy plant.

The powders were investigated using particle size analysis on OCCHIO 500 nano measuring equipment (OCCHIO, Angleur, Belgium) and using a Tescan Vega 3LMH scanning electron microscope with an EDS detector (Tescan, Brno, Czech Republic). The preparation of the powder composite material was carried out by mechanical alloying in a Retch PM 100 planetary ball mill (Retch, Haan, Germany).

Production of experimental samples was carried out on a special laboratory machine for selective laser melting—an ALAM (advanced laser additive machine)—that was developed and assembled at MSTU “STANKIN” (Moscow. Russia). The SLM process diagram is shown in Figure 1 [27].

The processing regimes were determined in several stages. The technique for achieving optimal regimes are described in detail in previous publications [18,19,20].

The main stages were: initial assessment of the optimal set of parameters using mathematical modeling; determination of processing modes in the manufacture of single tracks; definition of processing modes in the manufacture of single layers; and definition of processing modes in the manufacture of volumetric objects.

This method allows the exclusion of unsatisfactory modes and moving on to the next steps. Each step allows one to define a group of parameters for the objects: single tracks (laser power, scanning speed); single layers (hatch spacing); and volumetric objects (scanning strategy).

The powder layer thickness ranged from 10 to 100 μm. The powder layers were scanned by a laser beam with a wavelength of 1.07 μm, with a laser spot diameter of about 100 μm. The laser beam power varied from 30 to 180 W; the scanning speed was from 30 to 350 mm/s.

In previous works [7,18], it was established that when obtaining single layers, it is advisable to use a hatch spacing with a 30% overlap. In the manufacturing of volumetric objects in order to reduce the anisotropy of mechanical properties, the direction of scanning by laser must be turned from layer to layer by 900 [7]. In this regard, the multidirectional interlayer scanning strategy was used (Figure 2b). In addition, with a unidirectional scanning strategy, the product has more pores and greater deviation from shape than a product made by the multidirectional strategy.

Metallographic analysis of the obtained samples was carried out using optical and electron microscopy. The chemical composition was determined by energy dispersive X-ray spectroscopy using a Tescan VEGA 3 LMH scanning electron microscope (Tescan, Brno, Czech Republic). Diffraction spectra were obtained on a PANalytical Empyrean Series 2 X-ray diffractometer using CoKα radiation. The analysis of the phase identification was carried out using PANalytical High Score Plus software (3.0, Malvern Panalytical Ltd, Almelo, The Netherlands) and the ICCD PDF-2 database. The experiments to determine wear resistance under abrasive conditions were carried out on a Calowear Anton Paar machine. (Anton Paar GmbH, Graz, Austria) The counterbody was a hardened steel ball with a diameter d = 25.4 mm. An abrasive suspension was fed into the contact zone between the ball and the sample surface. We used 2 types of suspensions based on silicon carbide with a particle size of 2–5 μm and a diamond suspension with a crystal size of 9 μm. To determine the hardness of the samples, a Qness Q10A microhardness tester (Qness GmbH, Golling, Austria) was used. The fracture toughness coefficient was measured according to the method proposed in [16]

## 3. Results

### 3.1. Preparation and Study of Powder Compositions for SLM

When preparing the initial mixtures to obtain high wear resistance, we proceeded from the possibility of creating a mixture with a maximum amount of micron tungsten carbide provided that it was uniformly distributed in a nanostructured cobalt binder.

In preparing the initial mixtures for obtaining high strength properties, we proceeded from the possibility of creating a mixture with the maximum amount of tungsten carbide. When preparing powder mixtures of micron tungsten carbide and nanocobalt, it is necessary to take into account the surface areas of the powders. The surface area of the cobalt powder *S_co_* should be greater than or equal to the surface area of the tungsten carbide powder *S_wc_* for homogeneous coating by nanocobalt particles of tungsten carbide.

To determine the area of the components in a mixture of WC and Co, the shape of the powder particles was taken as a perfectly round ball. Then the surface area of one particle is equal to:
*S_particle_* = 4*πr*^2^(1)


With an average radius of the tungsten carbide particle *r_WC_* = 400 nm and an average radius of the cobalt particle *r_Co_* = 40 nm, the area of one particle *S_particle__Co_* = 20,096 nm^2^ and *S_particle__wc_* = 2,009,600 nm^2^. They are related as 1/100. Therefore, for homogeneous dusting of one WC particle, 100 particles of Co are needed. Knowing the volume and density of the chemical component, it is possible to find the mass of one particle:*M_particle_* = *ρV*(2)
where *V* is the volume of one particle:*V* = 4/3*πr*^3^.
(3)


Thus, the mass of one particle of cobalt powder *M_particle Co_* = 2,384,725.63 × 10^−27^ g, and the mass of the tungsten carbide particle *M_particle WC_* = 4,233,557,333.86 × 10^−27^ g. Therefore, in 100 grams of the 94 wt% WC and 6 wt% Co composition, one can determine the number of particles *n*:
*n* = *M_total_*/*M_particle_*.
(4)


The number of cobalt particles is *n_Co_* = 2.5 × 10^21^, and the number of tungsten carbide particles is *n_WC_* = 2.2 × 10^19^. Then the surface area of one component of the mixture *S_component_* is equal to:
*S_component_* = *n S_particle_*.
(5)


The above estimates indicate that in 100 grams of the composition, 94 wt% WC and 6 wt% Co, the surface area of the cobalt powder *S_Co_* = 50,240 × 10^3^ m^2^ is greater than the surface area of the tungsten carbide powder *S_WC_* = 44,211.2 × 10^3^ m^2^.

The powder is highly agglomerated, which is a consequence of its submicron size. The distribution of chemical elements is heterogeneous because cobalt nanopowder is agglomerated (Figure 3a).

In this regard, the powder mixture was processed in a ball mill. After processing, the powder mixture was also studied using SEM; as a result it was revealed that the treatment made it possible to achieve a homogeneous distribution of elements and break up the agglomerates (Figure 3b,c).

The investigated characteristics of the resulting powder mixture are summarized in Table 1.

A histogram of the particle size distribution of the powder is shown in Figure 4.

The measured average dimension for WC and Co was 800 nm and 80 nm, respectively (Figure 3c). The study of the chemical composition by energy dispersive analysis is shown in Figure 5. 

### 3.2. Structure and Properties of Samples Obtained by SLM

We studied samples obtained by SLM using the ALAM laboratory machine: single tracks, single layers, and volume cubes. Samples were obtained by varying the process parameters: powder layer thickness (H), laser radiation power (P), scanning speed (V), and hatch spacing (s).

Figure 6 shows the cross sections of typical samples of single tracks and single layers obtained by the SLM method. Samples were obtained by varying the process parameters. The parameters are indicated in the captions to the figures: H is the thickness of the powder layer, P is the laser radiation power, V is the scanning speed, s is the hatch spacing.

Based on the studies described above, a mixture with the maximum possible amount of tungsten carbide was selected under the condition of homogenous coating of micron tungsten carbide by nanocobalt: 94 wt% WC and 6 wt% Co.

A single track is an elementary unit in the method of selective laser melting. The SLM process can be considered as a complex of tracks for forming a layer and a complex of layers for forming a volumetric object. Therefore, the parameters were found, at which stable tracks were obtained. It will be rational parameters for obtaining a single layer.

The search for modes for obtaining stable single tracks was carried out according to three parameters: the powder layer thickness, the power of the laser, and the scanning speed. The range of variable parameters is indicated in the "Materials and Methods" section. Metallographic analysis of the cross section of the tracks showed that with the decreasing in the thickness of the applied layer, the geometry of the cross section of the track was more correct, while with a high parameter of the thickness of the layer, the track had a relief surface with metal growths at the borders. A thickness of 20 mm was chosen as the rational value of the thickness of the applied layer because applying a thinner layer is mechanically difficult by the leveling system of the machine, and the increased thickness requires more laser power or lower scanning speed, which leads to overheating and subsequent cracking. 

Varying the scanning speed at a constant power of 70 W showed that with an increase in speed, the heterogeneity of the structure decreased, and the geometry of the track became more stable (width of the track, height). In addition, increasing the scanning speed reduced the overheating of the material, reduced the probability of cracking, and increased productivity, but at speeds above 300 mm/s, the track did not have strong contact with the substrate.

Varying the laser power in the above ranges showed that the increase in power from 70 W to 150 W led to a significant decrease in the height of the track, which is explained by the evaporation of the material. It was experimentally revealed that the height of the track at a power of 150 W (V = 300 mm/s) was 4 times less than at a power of 70 W. At a power of less than 50 watts, the track had an islet shape and sometimes did not have contact with the substrate, which indicated insufficient energy to transfer material. In addition, an increase in power above 70 W led to the appearance of microcracks on the surface of the tracks (Figure 7), which when a monolayer was fabricated, transformed into large cracks passing through large sections (Figure 8).

Rational regimes were established for a mixture of 94 wt% WC and 6 wt% Co: scanning speed—300 mm/s, powder layer thickness—20 μm, laser power—50–70 W.

Fusion of a mixture of 94 wt% WC and 6 wt% Co with a high content of hardening carbide phase gave homogeneous formed tracks with strong metallurgical contact with the substrate (Figure 9).

This result was explained by the structure of the powder. Nanosized cobalt powder has an extended surface and, as a result, the melting temperature was lower than that of the micron powder, which made it possible to obtain a melting pool with a homogeneous distribution of unmelted WC micron particles.

For further studies of the structure and properties, volumetric samples were made according to the above optimal regimes.

We studied cross sections on an SEM. According to energy dispersive microanalysis, the quantitative and qualitative composition of the fused samples corresponded to the initial powder compositions. The research results are shown below in Figure 10 and Figure 11.

Metallographic studies confirmed that the obtained samples from WC–Co powder mixtures with a content of 94% Co formed a metallurgical contact with the substrate, and the distribution of chemical elements over the cross section was homogeneous. In general, compositional heterogeneities in the fused samples were much smaller than in the substrate. The scale of heterogeneity in the material obtained by the SLM was less than the resolution limit of the energy dispersive analysis used, which was about 1 μm. Therefore, we can say that the resulting material was homogeneous at a submicron level.

X-ray diffraction analysis of the fused powder mixture of 94 wt% WC and 6 wt% Co showed that the sample contained the main phases of W_2_C and WC. According to semi-quantitative analysis, the phase ratio was 90% W_2_C/7% WC; the rest was a cobalt-based solid solution. The X-ray diffraction pattern of this sample is shown in Figure 12.

Carbide W_2_C, which has a higher hardness than WC, prevailed in this composition, which suggested a high hardness of the material [28]. It was also found in [29] that the abrasion resistance of W_2_C material is two times higher than that of WC. 

A study of the influence of the percentage composition and size of WC fractions on the formation of cracks in parts made by SLM shows that when SLM is applied to brittle materials, the part often cracks already in the manufacturing process. In all cases, the fraction of the hardening phase without cracking in the metal matrix did not exceed 10%. This is not always enough to obtain the necessary mechanical properties. Attempts to increase the content of the hardening phase often lead to cracking of the material during SLM. The mixtures with different contents of WC were studied: 25%, 50%, and 94%.

For example, the composition of 50 wt% WC and 50 wt% Co (micron fraction of Co) without cracks in the manufacture of single tracks and single layers, cracked in the manufacture of a cube sample. Cracks formed on the surface of the layer and in the cross section of the material (Figure 13). 

Based on the experimental data, the accumulated experience on the SLM was confirmed—an increase in the content of the hardening phase led to cracking of the material. However, a composition containing 94% WC and 6% Co nanopowder was an exception to this statement; cracks were not observed either in single tracks, single layers, or in multilayer samples (Figure 14). 

Comparative tests on the wear resistance and hardness of a cube sample obtained by the SLM from a powder mixture of WC–Co 94%–6% showed that when using silicon carbide SiC as an abrasive, a recess did not form on the sample, which indicated a high hardness of the sample.

Tests for a sample of 94 wt% WC and 6 wt% Co were repeated with a diamond abrasive. To compare the coefficient of wear rate of the material, 94 wt% WC and 6 wt% Co, and carbide plates BK6 and BK20 obtained by conventional sintering were selected. The wear rate of the samples with a ratio of components of 94 wt% of WC and 6 wt% of Co was 1.3 times lower than that of a sample of BK6 obtained by the traditional method, with hardness 2500 HV_0.05_ to 1550 HV_0.05_, respectively. The values of the wear rate and microhardness are given in Table 2. The observed wear scars are shown in Figure 15. 

Fracture toughness tests were carried out on polished samples. The average crack resistance coefficients were calculated from three prints on each sample (Table 2). The results showed that 94 wt% WC and 6 wt% Co were more fragile than the BK6 and BK20 by 2 MPa∙m^1/2^ and 3.4 MPa∙m^1/2^, respectively, which, given the high hardness, is a good result.

## 4. Conclusions

Selective laser melting of metal-powder compositions of the WC–Co system was studied using nanosized powder of Co. The wear-resistant cermet material was obtained with a ratio of components: 94 wt% of WC and 6 wt% of Co. The rational conditions for the SLM were determined, which provided high hardness and wear resistance of the ceramic–metal composition. The structure of the obtained material consists of tungsten carbides W2C (90%), WC (7%), and FCC (face-centered cubic) solid solution based on cubic high temperature modification α-Co (Co_0.87_, W_0.13_) (3%). Due to the increased surface area, nanocobalt powder has a melting point lower than that of micron powder, which makes it possible to obtain a liquid melt pool with a homogeneous distribution of unmelted micron particles WC. The hardness of the obtained samples containing 94 wt% WC and 6 wt% Co is 2500 HV. The wear rate of the obtained samples is 1.3 times lower than that of a sample from BK6 obtained by the conventional method. The developed process parameters can be recommended for the manufacture of complex parts working in increased abrasive wear conditions.

## 5. Patents

As a result of work, 2 patents were obtained:
RU2669034C1. The method of obtaining products from powder material 94WC6Co.RU2669135C1. The method of manufacturing products by selective laser melting of a powder composition WC–Co.


## Figures and Tables

**Figure 1 materials-12-03425-f001:**
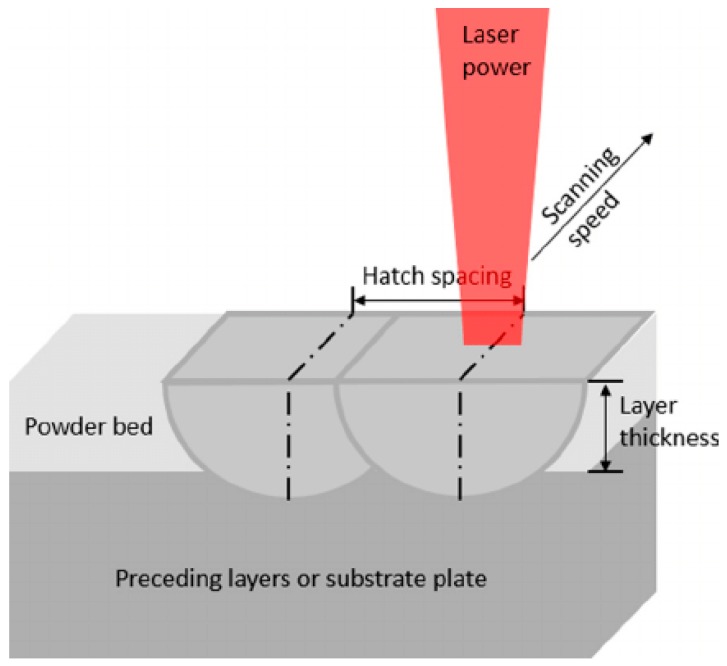
Schematic diagram of the selective laser melting (SLM) process.

**Figure 2 materials-12-03425-f002:**
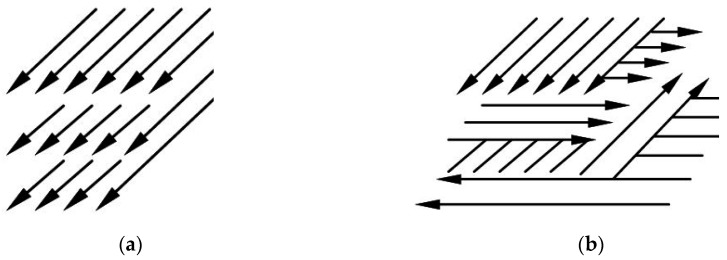
The scheme of the trajectory of the laser beam for various scanning strategies: (**a**) a unidirectional scanning strategy; (**b**) multidirectional scanning strategy.

**Figure 3 materials-12-03425-f003:**
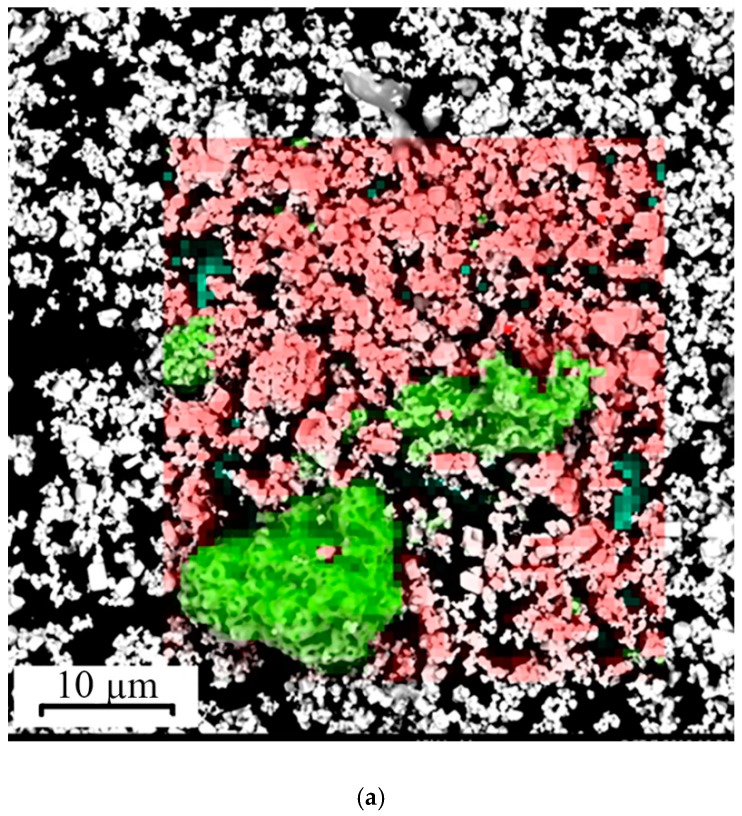
Photos of (**a**) the distribution of chemical elements in a mixture of 94 wt% micron powder tungsten carbide (WC) (red—tungsten) and 6 wt% nanopowder cobalt (Co) (green—cobalt) before processing in ball milling and (**b**) after processing in a ball mill. (**c**) SEM image of the mixture after processing in a ball mill.

**Figure 4 materials-12-03425-f004:**
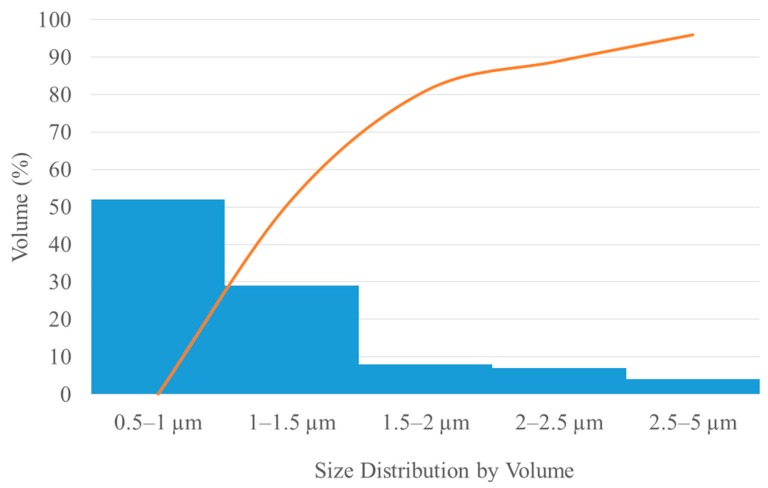
A histogram of the particle size distribution.

**Figure 5 materials-12-03425-f005:**
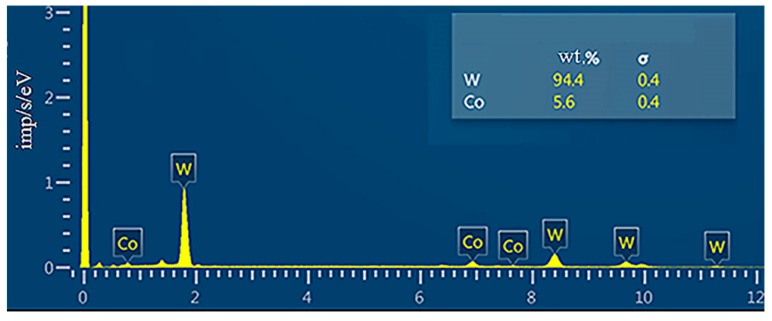
Energy dispersive spectrum of a mixture of micron powder WC and nanopowder Co.

**Figure 6 materials-12-03425-f006:**
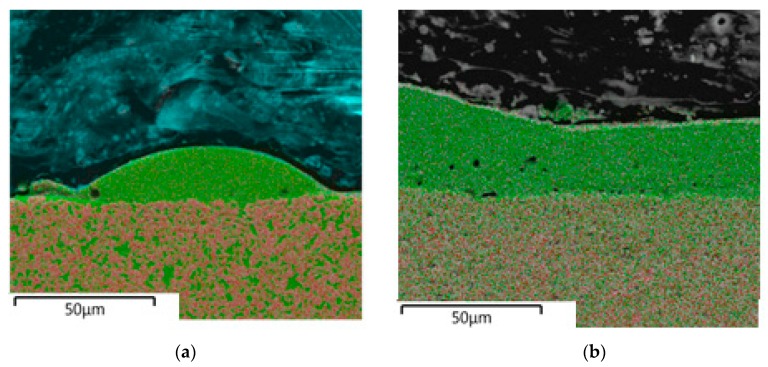
Cross sections of fused samples: (**a**) single track at H = 20 μm, P = 70 W, and V = 300 mm/s; (**b**) monolayer at H = 35 μm, P = 70 W, V = 300 mm/s, s = 50 μm.

**Figure 7 materials-12-03425-f007:**
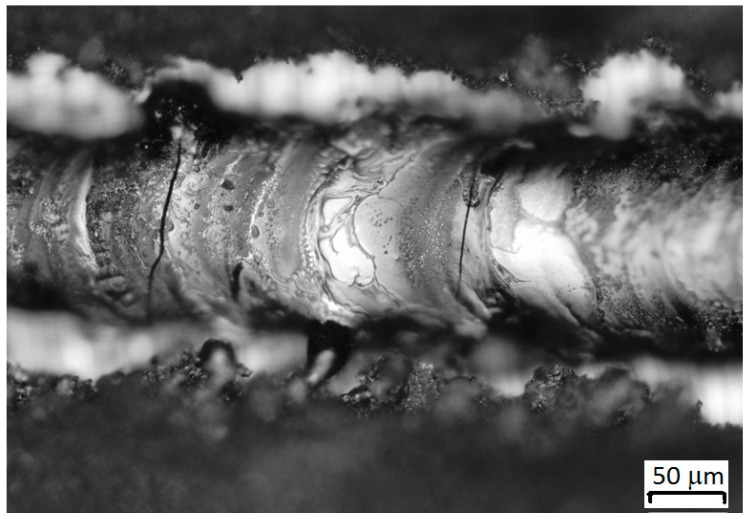
Cracks on the surface of a single track—94 wt% WC and 6 wt% Co P = 120 W, V = 300 mm/s.

**Figure 8 materials-12-03425-f008:**
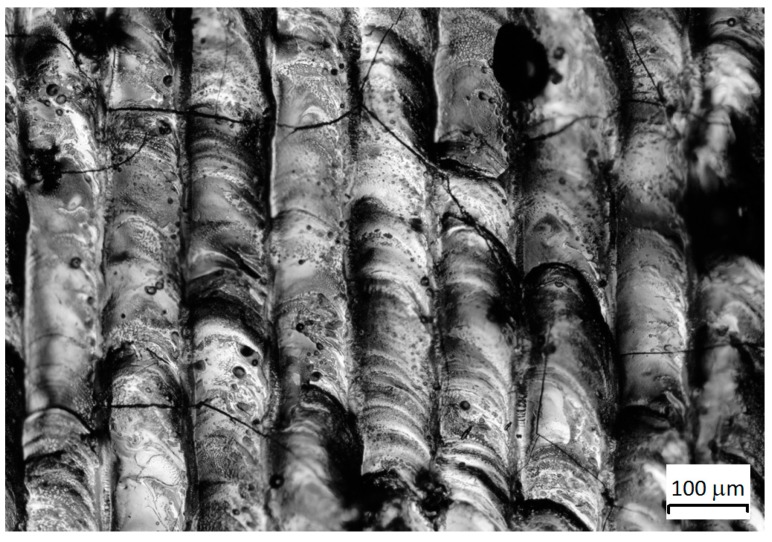
Cracks on the surface of the monolayer—94 wt% WC and 6 wt% Co P = 120 W, V = 300 mm/s.

**Figure 9 materials-12-03425-f009:**
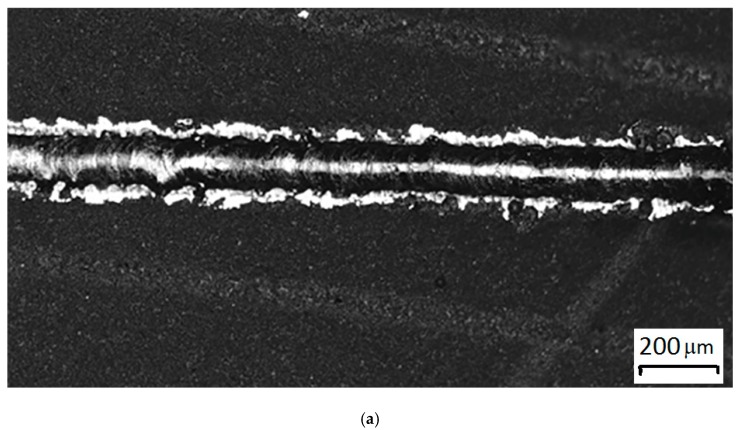
Samples on the substrate, BK20, hard alloy with a content of 20% Co: (**a**) single track of 94 wt% WC and 6 wt% Co, P = 70 W, V = 300 mm/s, N = 20 μm. (**b**) monolayer of 94 wt% WC and 6 wt% Co P = 70 BT, V = 300 mm/s, N = 20 μm, s = 70 μm.

**Figure 10 materials-12-03425-f010:**
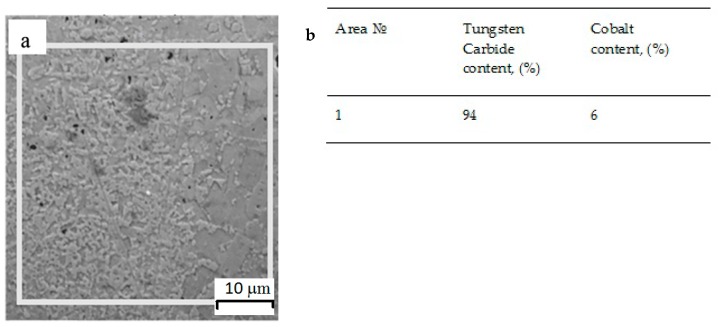
Spectral microanalysis of a cube sample of 94 wt% WC and 6 wt% Co; (**a**) study area, (**b**) percentage of elements.

**Figure 11 materials-12-03425-f011:**
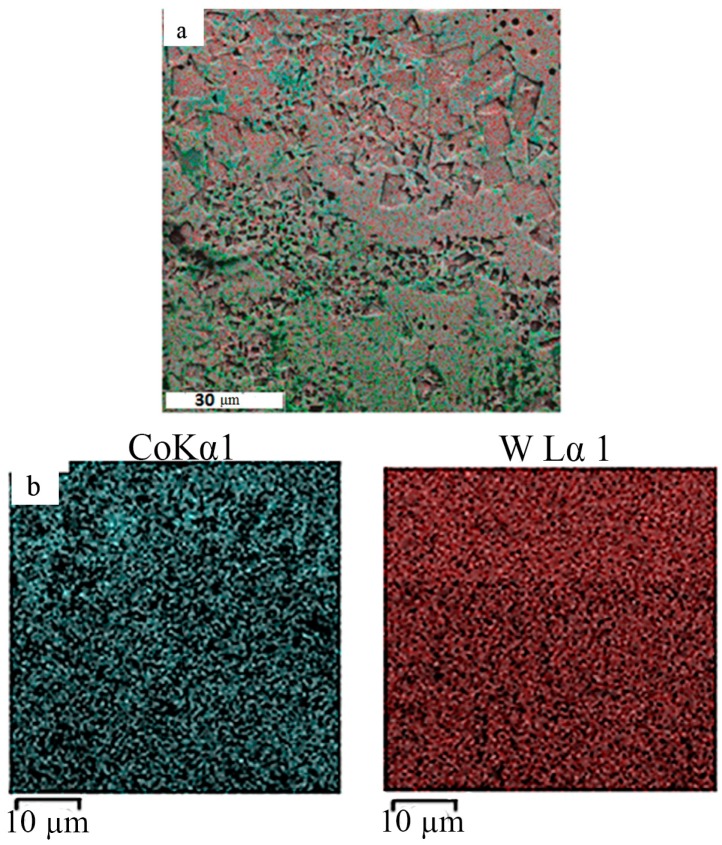
Distribution of chemical elements in a cube sample 94 wt% WC and 6 wt% Co; (**a**) total distribution, (**b**) distribution of individual elements, where the left figure is the distribution of cobalt (blue—cobalt) and the right figure is the distribution of tungsten (red—tungsten).

**Figure 12 materials-12-03425-f012:**
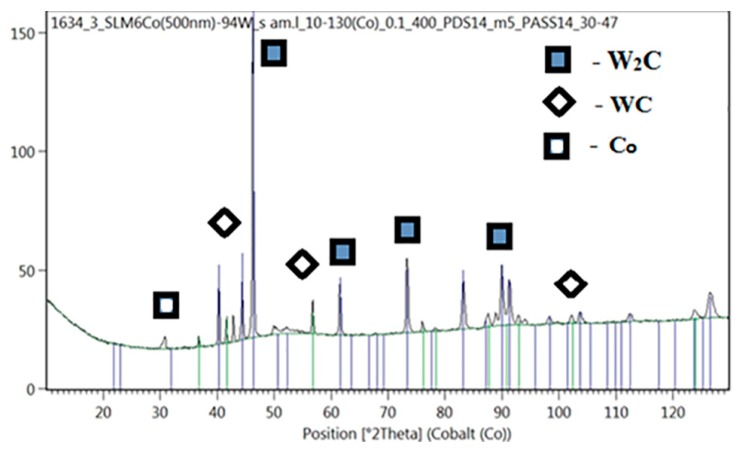
Diffraction spectrum of a sample of 94 wt% WC and 6 wt% Co.

**Figure 13 materials-12-03425-f013:**
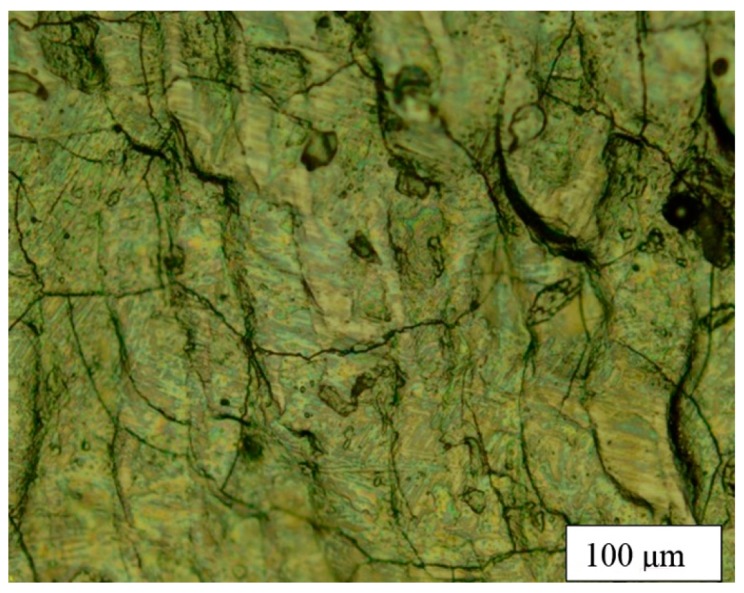
Cracks in the material 50 wt% WC and 50 wt% Co (micron fraction Co), top view.

**Figure 14 materials-12-03425-f014:**
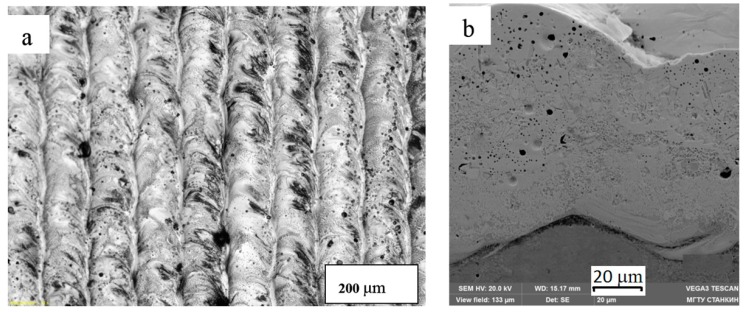
Multilayer WC–Co sample 94%–6%: (**a**) top view, (**b**) cross section.

**Figure 15 materials-12-03425-f015:**
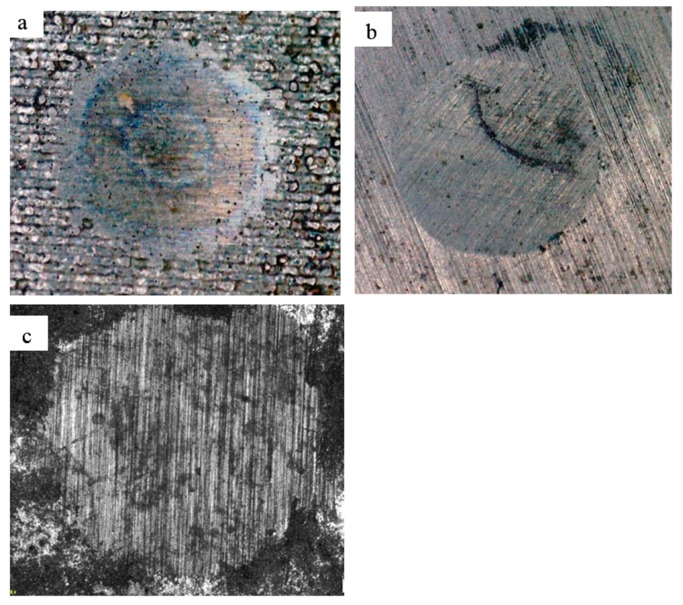
The observed wear scars on the surface of the tested samples: (**a**) 94 wt% WC and 6 wt% Co, (**b**) BK6, (**c**) BK20.

**Table 1 materials-12-03425-t001:** Characteristics of the finished mixture of micron WC and Co nanopowder.

Parameter	Value
Total particles investigated	216,452 pc
Granulometry	0.5–5 μm
Average particle size	0.8 μm
Form	irregular
Chemical composition	94% WC, 6% Co (trace Si, Cu, Ni, Fe, C, O)
Structure	cast material

**Table 2 materials-12-03425-t002:** The value of the wear rate for samples.

Product Material	94 wt% WC + 6 wt% Co	BK6	BK20
Wear rate × 10^−13^ (m^3^/m∙N)	8.4	11	43
Hardness (HV)	2500	1550	1050
Fracture toughness coefficient (MPa∙m^1/2^)	6.9	8.9	10.3

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
