# Peer review of "Possibilities of Manufacturing Products from Cermet Compositions Using Nanoscale Powders by Additive Manufacturing Methods"

_materials, 2019, doi:10.3390/ma12203425_

Round 1

Reviewer 1 Report

This paper demonstrates the improved material properties of a ceramic-metal composite fabricated by selective laser melting. The topic is of interests to the reader and the results appear to be promising. Here are some comments that the authors can consider to improve the paper.

The literature review on the relevant topic can be more comprehensive. The reviewer believes there are more related papers that should be included in the review. This would help to highlight the technical challenges that this paper is addressing. The materials and methods section is very short. I think it would be useful to include the methodology that the authors used to achieve the optimal regimes of parameters for the SLM process. What are the critical or most important process parameters? This would provide insight for the readers and this knowledge would be applicable to other composite as well. The authors touched on this in 3.2 but there’s no sufficient details are provided. Some more results can be included as well In the conclusion section, the authors had “relationships between the parameters of selective laser melting, the structure, and the physicomechanical properties of the samples are established.” It’s unclear what these relationships are and what results the authors had to support this. The authors need to provide more results from the optimization process. The conclusion section can be written in a more conventional format, instead of using bullet points.

Author Response

Please see the attachment. Comments at the end of the file

Reviewer 2 Report

The authors present tungsten carbide-cobalt composite obtained by the selective laser melting (SLM) technology. The subject and the SLM technology are interesting, and this paper can be published in Materials after minor changes. My questions and comments are as following:

Please explain all the abbreviations in the abstract; WC as tungsten carbide alloy, etc. Introduction: Line 72, the authors should be more specific with the aim of their study. All the methods and analysis techniques are mentioned vaguely later in the text, but I think more refined aims of this work should be proposed at this stage. Line 98: The authors wrote ‘In preparing the initial mixtures for obtaining high strength properties, we proceeded from the possibilities of creating a mixture with the maximum amount of tungsten carbide.‘ Can you reformulate it to explain your initial aim? …proceeding from the possibilities of creating a mixture.. does not make much sense to me. Line 124: How did the authors get the average particle size estimation in Fig.1? Is an image analysis program used? If so, it would be better to present a particle size distribution graph. If the white bar shown in the corner of the image is the scale bar, being 400 nm, it seems to me that the average particle size is smaller than 800 nm for WC. A similar approach should be taken for the sample after ball milling. It is not very convincing that ball milling made the powder more homogeneous. Cobalt remained still agglomerated as observed in the figure 4b, but he agglomerated Cobalt particle size seems somewhat smaller. To make this point clearer, the authors should choose zoomed-in images for Fig 2a and Fig3a. A scale of 50 microns is too large to understand if ball milling was useful. If Fig 1 to 3 is from the same batch of a sample, they can all be combined. Similarly, a zoomed-in SEM image should be added to Fig 4.  Line 146. The authors started a new section on the processing without giving initial technical information in section 2. Processing and fabrication parameters, processing strategies (single-track single layers, volume cubes, etc.). need to be detailed in Section 2. Also, a schematic of the SLM and/or the processing designs could be added here or if the authors previously published on this, a citation could be referred.  Line 151: which substrate is this? Line 158: Optimal parameters are given here, but the authors should write what parametric they changed. I believe the authors should show a collection of images and results according to the parametric study they performed, and then they should propose an optimal parameter. I cannot follow at which point the authors talk about the samples prepared by the different processing strategies: single track vs single layer vs cube. It is not clear in the figure captions and the text. The text kept jumping to a new figure before a point was made from the previous strategy. What are the advantages and disadvantages of each processing strategy? The explanation and the discussion need to be extended in the text. Line 219 and 231 and table 2: is this letter (и) supposed to be there? Generally, the scales in each figure is really small to read. If printed in a paper, it will not be visible to the reader. Line 20: C in Ceramic is bold. Line 54: space before a comma. Line 88: Phase identification rather than phase composition.

Author Response

(The authors gave the same response as above.)

Round 2

Reviewer 1 Report

The authors have addressed all the comments and the quality of the paper has improved.